# Effectiveness of a 12-Month Online Weight Reduction Program in Cohorts with Different Baseline BMI—A Prospective Cohort Study

**DOI:** 10.3390/nu14163281

**Published:** 2022-08-11

**Authors:** Jakub Woźniak, Katarzyna Garbacz, Olga Wojciechowska, Michał Wrzosek, Dariusz Włodarek

**Affiliations:** 1Department of Dietetics, Institute of Human Nutrition, Warsaw University of Life Sciences (WULS–SGGW), Nowoursynowska 159 C, 02-776 Warsaw, Poland; 2Centrum Respo, Chmielna 73, 00-801 Warsaw, Poland

**Keywords:** obesity, overweight, human, energy restriction, lose weight, BMI, lifestyle, online intervention

## Abstract

The purpose of this article was to answer the question of whether people who want to reduce their body weight can achieve different results depending on their baseline BMI and whether the rate of weight loss is constant over the months of intervention. The study included 400 individuals aged 19 to 55 years with a mean BMI of 31.83 ± 4.77 (min 25.1 max 51.8). Men comprised 190 subjects and women 210 subjects. The participants were divided into three groups with the following BMI: overweight, class 1 obesity, and obesity class > 1 (class 2 and 3 combined). BMI groups were randomized by gender, the number of trainings per week, training time, intervention length, and intervention type. The online intervention consisted of a 15% energy deficit diet and training. Over the 12-month dietary intervention, overweight subjects reduced average body weight by 16.6%. The group with class 1 obesity reduced body weight by 15.7%. The group with obesity class > 1 reduced mean body weight by 15.4%. The relative weight reduction in the overweight group was significantly greater than in the other obesity groups (*p* = 0.007). In all groups, the rate of weight loss from month-to-month was statistically significant (*p* = 0.0001), ranging between 0.6 and 2.6% per month. The results indicate that overweight individuals are likely to experience a percent greater weight loss as a result of a comprehensive lifestyle intervention. Regardless of baseline BMI, the observed weight loss was consistent from month-to-month throughout the 12-month period, which may indicate that the diet, as well as, the training plan were properly tailored to the subjects’ needs and that they were highly motivated to participate in the program throughout its course. Properly conducted lifestyle intervention enables significant weight loss regardless of baseline BMI values.

## 1. Introduction

For decades, the epidemiology of obesity has been evolving significantly. It is estimated that about 53% of the population in the EU is considered above the normal weight (36% overweight and 17% obese) [1]. Obesity is positively correlated with an increased risk of diabetes, cardiovascular diseases, musculoskeletal disorders, chronic kidney disease and several types of cancer [2]. Moreover, the recent COVID-19 outbreak appears to be another disease exacerbated by obesity [3]. In addition, the use of prolonged lockdown as a way of combating the pandemic might contribute to the increased percentage of obese individuals globally. Furthermore, increased stress levels and decreased physical activity due to the closure of numerous sports facilities could lead to an excessive energy intake [4].

In addition, due to the COVID-19 pandemic and a global necessity to impose lockdowns, there has been a need to introduce a different way of dietary consultations and education. Thanks to the expeditious development of web-based applications and the shift to a remote model of healthcare delivery, online dietary services were made possible [5]. Moreover, due to a more sedentary lifestyle because of lockdowns, restricted rules of spending time outside and the closure of indoor sports facilities, an increase in body weight among individuals worldwide has been observed [6]. In recent years, there has been an interesting shift in the research in the field of online reduction programs for people with obesity. A large, a non-randomized trial was carried out to compare results from online diabetes prevention programs and outcomes among individuals with in-person interventions [7]. The main outcome measure in the trial was weight change at 6 and 12 months. Two hundred and sixty-eight prediabetic individuals with excessive weight were enrolled in the online program, while 273 participants were included in the in-person scheme. Participation was evaluated based on the completion of weekly modules. In the first 6 months of the trial, 87% of patients completed eight or more modules of the online program, while only 59% of the in-person individuals. The results also showed that there was no significant difference in weight loss between online and in-person programs. The trial provides an argument that the online program has higher participation than in-person visits with no significant change in the effectiveness of weight loss. Moreover, a meta-analysis of 25 randomized control trials was carried out to establish the effectiveness of online dietary intervention in individuals with obesity and hypertension and/or type 2 diabetes [8]. The results provided evidence that the online dietary counseling was successful in significantly decreasing the weight of participants compared to the control groups. However, it was also suggested that the online intervention should be longer than six months for a better outcome.

It is crucial to acknowledge the importance of physical activity in obese or overweight individuals. In a quasi-experimental, quantitative, and longitudinal study, changes in lipoinflammation markers after a concurrent training program were demonstrated [9]. Twenty-six obese individuals were enrolled in an 8-week training regimen, which was a combination of aerobic and resistance exercise. The applied physical activity resulted in significant differences in the adiponectin–leptin ratio. The outcome from blood samples can be reflected in a decreased risk of cardiovascular diseases and lipoinflammation. Moreover, a systematic review of exercise programs in obese patients undergoing bariatric surgery was performed [10]. The results suggested a positive effect of physical activity on weight loss, muscle strength, percentage of fat-free mass cardiorespiratory endurance, and the overall quality of life.

What is also worth mentioning is the significance of mental health and therapy for obese individuals during weight loss programs. In the review, researchers combined intensive behavioral therapy benefits with diabetes prevention programs in obese patients [11]. The interventionists received training on how to talk about obesity sensitively and respectfully, as well as guidance about goal setting, problem-solving and other behavioral strategies. After 24 weeks of interventions, 46% of obese individuals lost ≥5% of their baseline weight, with an overall mean of 5.4%. After 1 year, 44% of participants lost ≥5% of their baseline weight, while the mean was 6.1%. The results provide the suggestion that behavioral therapy might be helpful during weight reduction programs. However, further research is needed to establish its specific beneficial effect on the weight loss regimen.

The most common method of determining nutritional status in adults is body mass index (BMI). It assesses people into categories by the person’s weight in kilograms divided by the square of the person’s height in meters (kg/m^2^) [2].

BMI is considered a simple and the most cost-effective method for obesity control at the population level. It is used to track the changes in the overall health status of the population, as well as goals or risks [12]. Nevertheless, the index has its drawbacks. It is still an indirect measure. It depends only on height and weight and does not include specifics of an individual’s body composition. BMI does not include adaptations that occur during the aging process, as the proportion between fat mass and free fat mass and bone mass changes [13]. However, the index can still provide helpful insights into obesity as an individual and social issue. It is widely used by government bodies, non-profit organizations, researchers and health professionals. BMI is considered to be the cheapest and simplest tool for tracking obesity levels among the population. Moreover, the definition and equation of BMI are available across different forms of media. That may encourage individuals to understand the issue of obesity as well as track their own health and nutritional status.

It is crucial to reduce the prevalence of obesity, not only for individual health improvement but also to decrease social costs. It is estimated that moderate weight loss of 5–10% is sufficient for health improvement [14]. Researchers and clinicians should target health promotion and decrease in risk of comorbidities rather than an exact number on a scale. An individually prescribed health goal should be acknowledged rather than a specific weight [15]. Therefore, baseline weight needs to be considered and recognized when weight-loss interventions are introduced. Furthermore, the goal, effectiveness and duration of the weight reduction intervention might differ in various groups of people classified by their BMI. However, the 12-month intervention seemed to be optimal in this specific online program. The design of the study was based on the results of the meta-analysis of 25 randomized control trials described before in the introduction. The researchers suggested that the online intervention should be longer than six months for better outcomes [8]. The field of the impact of baseline BMI on weight loss is not yet fully discovered.

This article aims to answer the question of whether people who want to reduce their body weight over 12 months can achieve different results depending on their baseline BMI and whether the rate of weight loss is constant over the course of individual months. For this purpose, we focused on expressing weight loss in kilograms as the absolute values, and as a percentage of the initial body weight, as the relative values. We prioritized three aspects related to reducing one’s body weight: to bring the individual’s body weight to a normal BMI, to reduce one’s BMI grade by 1 baseline degree and to determine the number of people who managed to reduce their body weight by at least 5%, 10%, 15% or 20% of their baseline BMI.

## 2. Methodology

### 2.1. Study Design

This study was designed as an observational, prospective, open-labeled, twelve-month trial. Data collected concerned the weight loss program from January 2019 to December 2021.

### 2.2. Sample

A total of 400 subjects who participated for 12 months in a weight reduction program were included in the evaluation. In total, 720 people were included in the observation. In the course of the intervention, 320 people did not complete the entire program (44.44%). In the overweight group, there were 105 people. In the group with 1st class obesity, there were 111 people, and in the group with obesity > 1, there were 104 people. The subjects were aged between 19 and 55 years with a mean BMI of 31.83 ± 4.77 (min 25.1 max 51.8). Men constituted 190 subjects and women 210 subjects. Inclusion criteria were as follows: age 18–55 years, excessive weight (BMI > 24.99 kg/m^2^), no dietary interventions in the 24 months prior to entering the weight reduction program, participation in a dietary intervention for 12 months, no musculoskeletal injuries, accessibility to a computer and/or telephone, no physician contraindication to regular physical activity.

### 2.3. Study Procedure and Intervention Characteristics

The analysis included 400 individuals who completed a 12-month cooperation with a dietitian using the Respo method based on 4 pillars.

The Respo method (from the word “responsiveness”) is based on individual adjustment of the whole plan of losing weight, body shaping, and achieving better health and well-being specific to the needs of each patient. The dietitian selects the method of action so that it is maximally adapted to the patient. The time needed to prepare meals for the whole day is consulted with each client and chosen in order to make it easier and more likely for them to follow the diet plan for a long time. Meal times and quantities were not rigidly set by the dietitian but responsively negotiated with the patient. Trainings were adjusted to the patient’s fitness level, availability of training equipment, and the time they could devote to physical activity at home. Within the individually planned training program, each training was accompanied by an instructional video available to patients at any time during the cooperation. Patients reported completing the training as well as eating the meals each day of the cooperation. Most participants trained for 60 min, 3 times per week. Trainings were based on multi-joint exercises of the whole body adjusted to the level of training and abilities of patients. The third pillar of the method is support, which consists of constant online contact with a dietitian via online chat. To increase the possibility of contact with the dietitian, the patients could write to them from any electronic device with access to the internet. The last pillar of the method is the development of new, correct habits during the entire period of cooperation. For this purpose, there is a nutritional education adjusted to the realities of the patient’s everyday life.

Patients completed nutrition and medical forms prior to nutritional intervention. Anthropometric measurements were taken by patients at baseline and during the intervention [16].

The diet was designed according to the recommendations for healthy adults [17]. The energy value of the diet was reduced by 15% in relation to the body requirements, which was determined on the basis of the basal metabolism estimated using the Harris Benedict formula, including the physical activity index (PAL) according to the recommendations of the Food and Nutrition Institute [18]. The level of physical activity was assessed based on the physical activity questionnaire published by Johansson and Westerterp [19]. During the nutritional intervention, the energy value of the diet could be adjusted if no decrease or increase in body weight was observed over a 4-week period. The first action if the therapeutic intervention did not work was to monitor the patient’s adherence to the diet and physical activity guidelines. Details of the protocol for correcting the energy value of the diet and the amount of physical activity are described in Figure 1.

The PAL coefficient oscillated between 1.2 and 2. The proportion of carbohydrates in the diet was set at 50–55% of the energy value of the diet, with sugars added to 10%. The proportion of energy from fat was 25–35% of the energy value of the diet, and the protein supply was set at 1.6g per kg of body weight. The supply of vitamins and minerals was realized based on the standards for the population of healthy adults [18]. Study participants received a 7-day menu individually adjusted to taste preferences, which was modified over time while maintaining the dietary guidelines to minimize the participant’s desire to drop out of the study. The diet was balanced based on a computer program having a database of products and foods from the Food and Nutrition Institute and the USDA. All subjects were supplemented with 2000 IU of vitamin D during the period of participation. Table 1 shows the characteristics of the interventional diet.

During the dietary intervention, participants were in constant contact with the dietitian and trainer and submitted bi-weekly reports including body weight. All participants in the study had contact with a dietitian at least every 2 weeks to ensure the same model of intervention for each person. In addition, they completed a running diary each day to monitor the actual intake of the intended diet. Each study participant received appropriate instructions for completing the food diary. The diary included all foods and beverages consumed, expressed in household measures (e.g., glasses, spoons) and/or units of weight (g) along with the time of consumption. The food diary also included the supplementation used along with the type of formula and the daily dose. Any deviations from the protocol were immediately corrected through nutrition education, and the menu was modified while maintaining the dietary guidelines to minimize the desire to abandon the nutritional intervention.

### 2.4. Outcome Measurements

Height and weight measurements were taken by the subjects themselves after a pre-prepared training session in which a specialized dietician showed how to perform these steps correctly at home. Body height was measured at the beginning of the study by the subjects after careful instructions were provided by the dietician.

Body weight measurements were taken at the beginning of the study and every 14 days. Before each weight measurement, the patient was required to be familiar with the protocol for this activity, which included the need to place the scale on a flat, hard surface in the same location throughout the intervention period. Measurement took place while fasting in underwear. Each participant used a standardized balance from a manufacturer certified by the Central Office of Measures in Poland to ensure the accuracy of the measurements.

BMI was used to assess body weight. The classification was adopted from the WHO [2]:

Normal weight (BMI 18.5–24.9 kg/m^2^); overweight (BMI 25.0–29.9 kg/m^2^), obesity class 1 (BMI 30.0–34.9 kg/m^2^), obesity class 2 (BMI 35.0–39.9 kg/m^2^), and obesity class 3 (BMI > 40.0 kg/m^2^). Because few subjects had class 3 obesity for analysis, they were included in the group of subjects with class 2 obesity (this group was described as obesity class > 1). The number of overweight subjects was 161, with class 1 obesity was 135, and with obesity class > 1 was 104.

### 2.5. Statistical Analysis

The data obtained from the observations were collated and systematized using Excel spreadsheet tools. These tools were also used to calculate the derived parameters—BMI, change over time.

The quantitative study was conducted using the STATISTICA 13.3 PL package (TIBCO Software Inc. Warsaw, Poland (2017). Statistica (data analysis software system, Warsaw, Poland), version 13. http://statistica.io (accessed on 10 May 2022). For the entire statistical test, a level of *p* < 0.05 was taken as the cut-off for the rejection of the null hypothesis. Basic descriptive statistics for quantitative data were calculated, and distributions of qualitative characteristics were determined using multivariate (contingency) tables. The significance of differences in the distribution of qualitative characteristics was tested using the chi-squared test in combination with the multivariate tables. Due to the rejection for most of the analyzed variables by the W Shapiro–Wilk test of the hypothesis of normality of distribution and the expression of a significant part of the variables in ordinal scales, non-parametric tests were used in the study—the Mann–Whitney U test (with correction for continuity), the Wilcoxon signed-rank test, and ANOVA–Kruskal–Wallis or Friedman with post hoc tests. Student’s *t*-test for dependent groups was used to assess the differences in body weight changes over time.

## 3. Results

The number of overweight subjects, i.e., with BMI between 25 and 29.9, accounted for 161, those with class 1 obesity, i.e., with BMI between 30 and 34.9, accounted for 135, and those with class 2 obesity and extreme obesity, i.e., with BMI above 35, accounted for 104. Immediately before the nutritional intervention, the mean BMI of the subjects participating in the nutritional program was 31.83 ± 4.77 kg/m^2^. The mean age was 33.42 ± 7.2 years with a height of 1.73 ± 0.09 m. Physical activity as measured by the PAL index was moderate and amounted to 1.49 ± 0.15. More detailed general characteristics of the subjects are presented in Table 2.

The numbers of subjects after splitting into groups were as follows: 161 subjects for the overweight group, 135 subjects for the class 1 obesity group, and 104 subjects for the class 2 and 3 obesity groups. All BMI groups contained different amounts of men and women, but the difference was not statistically significant (*p =* 0.71). Overweight patients were significantly younger (31.9 ± 3.0 years) than obese patients (obesity class 1 and obesity class > 1: 34.2 ± 7.2 and 34.7 ± 7.7 years, respectively). All groups were characterized by similar height, number of trainings performed per week, time per training unit, and time spent training per week. The mean BMI in the overweight group was 27.6 ± 1.28, in the class 1 obesity group it was 31.8 ± 1.1, and the class > 1 obesity group it was 38.4 ± 3.5. The overweight subjects showed significantly higher (*p =* 0.0001) physical activity level (PAL), which was 1.57 ± 0.11 in this group. In contrast, both obese groups showed a lower PAL of 1.46 ± 0.13 and 1.39 ± 0.12, respectively. More relevant information characterizing the observed subjects is presented in Table 3.

Over the course of the 12-month dietary intervention, the overweight subjects reduced their mean body weight from 83.9 ± 10.6 to 70.1 ± 10.2 kg, which was associated with a weight reduction of 16.6%. From month-to-month, each weight loss that the group recorded was statistically significant (*p =* 0.0001) and was between 2.1% and 1.1% in each month, which in absolute measures was between 1.7 and 0.6kg. The group with class 1 obesity reduced weight from an average of 96.1 ± 11.1kg to 81.1 ± 12.4kg, which was associated with a relative weight reduction of 15.7%. From month-to-month, the weight reduction that was found in the group was also statistically significant (*p =* 0.0001) and was between 1.9 and 0.5% in each month, which in absolute measures, was between 2.2 and 0.4 kg. The final group with class > 1 obesity demonstrated a reduction in mean body weight from 114.5 ± 14.6 to 96.8 ± 14.6 kg, which was associated with a relative weight reduction of 15.4%. The month-to-month decreases in weight reduction that were observed in the group were statistically significant (*p =* 0.0001) at between 2.3% and 0.9% each month, which in absolute measures was between 3.0 and 0.8kg. Significant differences in the rate of weight loss in each month between groups were noted in months 2, 6, 8, 9, 10, 11 and 12 of participation in the program—the differences between the groups are shown in Table 4. To summarize the 12 months of the program, all three groups significantly reduced their body weight. In absolute values, the greatest reduction in body weight in kilograms was observed in the group with obesity class > 1, the mean value of reduced body weight was 17.7 kg compared to the group with overweight (13.8 kg) and the group with obesity class (1–15 kg) (*p =* 0.001). However, in relative terms, i.e., reduced percentage of body weight, a more significant reduction (*p =* 0.007) was found by those in the overweight group (16.6%). The group with class 1 obesity reduced body weight by 15.7%, and the group with class > 1 obesity by 15.4% (these values were not significantly different). More information about the changes in body weight in all three BMI groups is presented in Table 4.

The decrease in BMI in all three groups over each month was statistically significant (*p =* 0.0001). After 12 months, BMI in the overweight group decreased from a mean of 27.6 ± 1.28 to 23 ± 0.9, in the group with class 1 obesity, BMI decreased from a mean of 31.8 ± 1.1 to 26.8 ± 0.8, and in the group with class > 1 obesity, it decreased from a mean of 38.4 ± 3.5 to 32.5 ± 2.9. The rate of weight loss is shown in Figure 2.

Additionally, in relative terms, i.e., reduced percentage of BMI, a more significant reduction (*p =* 0.007) was found in the overweight group (16.6%). The group with class 1 obesity reduced BMI by 15.7%, and the group with class > 1 obesity by 15.4% (these values were not significantly different). The t-test results indicated significant differences between the body weight of the participants in the three-month intervals, between the start and the 3rd month (−5.03 ± 2.44 kg), the 3rd month and the 6th month (−4.18 ± 1.81 kg), the 6th month and the 9th month (−3.43 ± 1.5 kg), 9th month and the 12th month (−2.6 ± 2.15 kg) and between the start and 12th month (−15.27 ± 5.11 kg) (*p* < 0.0001 for all periods). These differences were also statistically significant in the individual BMI groups. More information about the changes in BMI in every month is presented in Figure 3.

In order to evaluate the effectiveness of weight reduction in each BMI group, we assumed the attainment of normal weight, a reduction in body weight by one baseline degree, and a relative weight reduction expressed as a percentage in order to qualify individuals as having achieved a weight reduction effect at a certain level. The first way was to determine how many individuals of each BMI group reduced their weight to a normal BMI, i.e., less than 25. In the overweight group, the final normal BMI was achieved by 86.3% of individuals; in the group with class 1 obesity, it was achieved by 11.9%; in the group with obesity class > 1, it was achieved by 0.9% of individuals. A second way to determine the success of the intervention was to reduce one’s BMI by one baseline degree. In the overweight group, this effect was demonstrated by 86.3% of individuals, in the class 1 obesity group by 99.2% of individuals, and in the class > 1 obesity group by 78.8% of individuals. The other four ways to determine success in weight reduction were to reduce baseline BMI by at least 5%, 10%, 15% and 20%. All participants achieved at least a 5% reduction in BMI, and almost all achieved at least a 10% reduction. More participants in the overweight group achieved at least a 15% and 20% reduction in BMI compared to the other two groups (Table 5).

The energy requirements of the observed patients also changed significantly (*p =* 0.0001) over the 12 months. Total energy requirements in the overweight group decreased from an average of 2741 ± 422 to 2350 ± 186.8 kcal, a change of 13.4%. In the group with class 1 obesity, the mean TMR decreased from 2867 ± 407 kcal to 2322 ± 205 kcal—this change was 18.9 %. In the group with >1-degree obesity, the mean TMR decreased from 3127 ± 462 to 2414 ± 202.9 kcal—this change was 22.7%. More data on changes in TMR, including changes in basal metabolic rate, are shown in Table 6.

## 4. Discussion

Analyzing the results, it is worth noting that the overweight subjects were significantly younger than the subjects in the other groups, which theoretically could have affected the effect obtained and could have been a differentiating factor in the weight loss results. Therefore, we additionally performed an analysis of weight loss in relation to age, and it showed no significant differences (*p =* 0.14) in weight loss due to the age of the observed subjects. It is noteworthy that compared to obese subjects, the overweight group also showed a higher leisure-time physical activity rate (PAL of 1.57 ± 0.11 vs. 1.46 ± 0.13 and 1.39 ± 0.12), and this may have influenced the final outcome in weight loss. Although the BMI groups had similar exercise-related activity, it is undoubtedly the case that greater spontaneous physical activity (NEAT) was a factor supporting weight loss in the overweight group.

Interesting results were observed by researchers Bautista-Castaño et al. [20]. In their study, a group of overweight patients also recorded higher weight loss. Although, all groups had similar levels of PAL. Additionally, the poorest outcomes applied to those subjects with childhood obesity and those who had obese parents. In another intervention-oriented study aimed at reducing patient weight, flexible endoscopic suturing for endoluminal gastric volume reduction was used. A multidisciplinary team provided post-procedure care. Patient outcomes were recorded one year after the procedure. The number of dietary and psychological contacts was a predictor of good weight loss results. In the linear regression analysis, adjusted for initial BMI, variables associated with %TBWL included frequency of dietary (β = 0.563, *p* = 0.014) and psychological contacts (β = 0.727, *p* = 0.025) [21].

Leaving aside leisure time activity, all groups in our study were similar in the number of trainings per week and the timing of these trainings, and thus, over the weeks, all individuals devoted a similar amount of time to training activity. Another strength of the study is the similar proportion of men and women in the three study groups. According to our literature analysis, there are very few publications dealing with the topic of the association of baseline BMI values with the rate of weight loss, and the available publications discussed below describe an intervention in which the majority of the study group was female. In a similar study to ours by Acharya et al. [22], the number of men was 14%, and in the study by Heshka et al. [23], men made up 18% of the group. This excluded the possibility of differentiation within groups due to the quantitative advantage of either sex.

What is worth emphasizing is that in our study, in all groups with different BMI values, a decrease in body weight was observed in each month of the program. There was no stopping of weight loss at any period of the intervention, which may indicate an appropriate adjustment of the diet and training plan for the subjects and their high motivation to participate in the program throughout its length. The rate of weight loss should be considered high compared to other similar studies analyzing this aspect. Additionally, it can be observed that the decrease in body weight in each three-month interval was decreasing during the 12–month intervention. The changes in body weight were most significant in the first three months and the least between the 9th and 12th months (measured in kilograms). In another intervention study based on lifestyle changes, a weight loss of 2.3% for men and 1.6% for women was reported over 12 months [24]. In the study by Sacks et al., weight loss after 12 months was 7% of the original body weight. Participants, similarly to our study, were subjected to a nutritional intervention combined with physical activity for 90 min per week [25]. Comparable results were also reported by Reseland et al. [26]. In this 12-month intervention involving an energy deficit diet and exercise program, subjects reduced their body weight by an average of 6.8%. Our study shows that regardless of their baseline BMI, the observed subjects achieved great success in reducing their body weight, but the effect is more pronounced in overweight subjects. These findings stand in opposition to the results of literature reviews on whether BMI is a differentiating factor in the rate of weight loss following a lifestyle change intervention. A 2014 systematic literature review bringing together the analysis of 13 intervention studies indicates that there were no differences in percentage weight loss among groups with different BMIs [27]. Similar findings were noted by the authors of a literature review on this topic from 2005 [28]. However, it is worth emphasizing the fact that the studies analyzed in these systematic reviews had different methodologies and differed from each other. Some of them were based only on the change in diet, others on dietary recommendations, and still, others were based on the change in diet and increasing the level of physical activity.

In our study, we sought to determine what percentage of observed individuals achieved success as a result of the intervention. We chose to focus on three aspects related to reducing one’s body weight. The first and most rigorous was to bring one’s body weight to a normal BMI. In our study, the overweight group overwhelmingly achieved this goal (86.3%). Significantly less of those with class 1, 2 and 3 obesity reduced their body weight below a BMI of 25. The second established determinant of weight loss success may be reducing one’s BMI grade by one baseline degree. Given this factor, by far, the majority of each BMI group achieved their goal. Our last suggestion for evaluating weight loss success is to determine the number of people who managed to reduce their body weight by at least 5%, 10%, 15% or 20% of their baseline BMI, which on an ascending scale shows how much weight loss people in each BMI class can count on. Here, too, according to our observations, individuals in the overweight group reduced their body weight to the greatest extent. Finally, it is also worth noting that as body weight decreases, our basal and total energy requirements decrease [29,30]. Therefore, after a successful weight loss intervention, further nutritional education for patients is necessary to teach them how to eat in order to maintain their lower body weight.

### Limitations of the Study

The study had some limitations. Due to the nature of the online intervention, we cannot confirm with certainty that the subjects adhered to the protocol 100 percent. Of course, in the course of the intervention, we checked the degree of program implementation. The study lasted 12 months, but it did not assess the degree of weight maintenance after the intervention. Due to the technological limitation, we also did not measure energy expenditure during exercise and throughout the day, which could have influenced the observed results. Furthermore, the study was unable to compare the mobile-based lifestyle intervention group with the offline intervention group to tell whether the online intervention was more effective than the offline one.

## 5. Conclusions

### 5.1. Main Conclusions

In conclusion, applying the same lifestyle changes (including changes in physical activity and diet) to individuals with different initial BMI can result in different outcomes in particular BMI classes. It is likely that the greatest effects will be achieved by overweight individuals, although this hypothesis would still need to be confirmed in other scientific studies.

An online weight reduction program with close follow-up and a program using dietary modifications and recommendations for physical exercise is an effective method to reduce weight in overweight and obese patients. Reduction in body weight is higher in obesity class > 1 in comparison with obesity class 1 and overweight patients, and % of weight loss is higher in overweight in comparison with obese patients.

Subjects with lower BMI achieved more significant reductions in their relative body weight than obese ones. In our study, the overweight participants had slightly higher non-training physical activity, which could have influenced the results.

### 5.2. Other Conlusions

A very important observational result is that the weight loss in all groups was almost linear, and all subjects were successful in weight reduction. Almost all the participants had at least a 5–10% decrease in BMI. At least a 15% decrease in BMI was observed most often with the overweight participants. Factors that most likely influenced this were the appropriately selected dietary and exercise intervention, patient support throughout the study period, and adequate levels of motivation within the patient group. Diet individually adjusted to the patient’s taste preferences increased the success rate of the plan.

## Figures and Tables

**Figure 1 nutrients-14-03281-f001:**
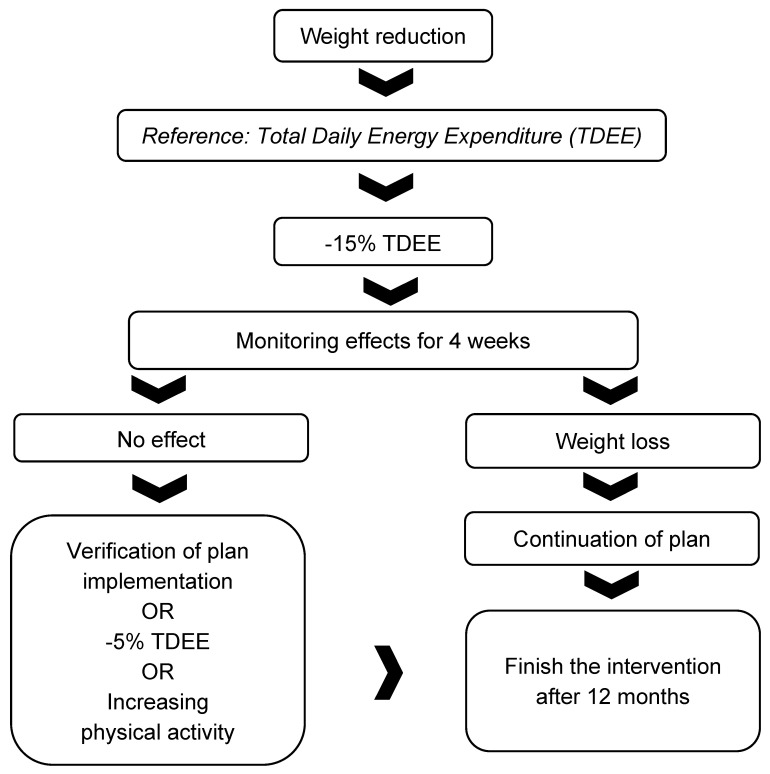
Protocol for adjusting the energy value of the diet throughout the nutrition intervention.

**Figure 2 nutrients-14-03281-f002:**
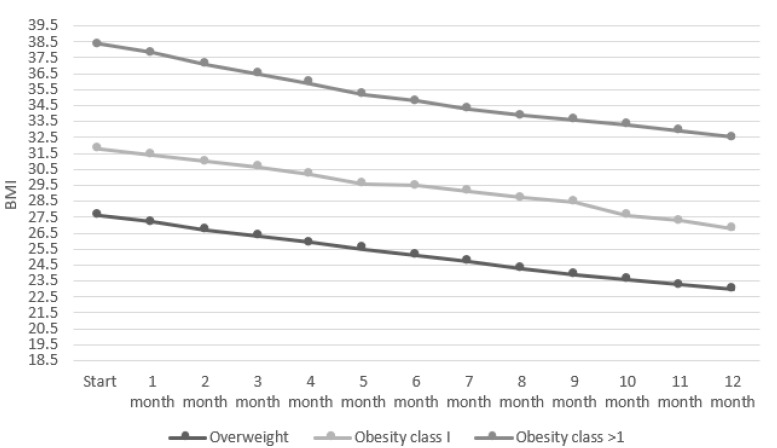
Rate of decline in BMI over 12 months. BMI: body mass index.

**Figure 3 nutrients-14-03281-f003:**
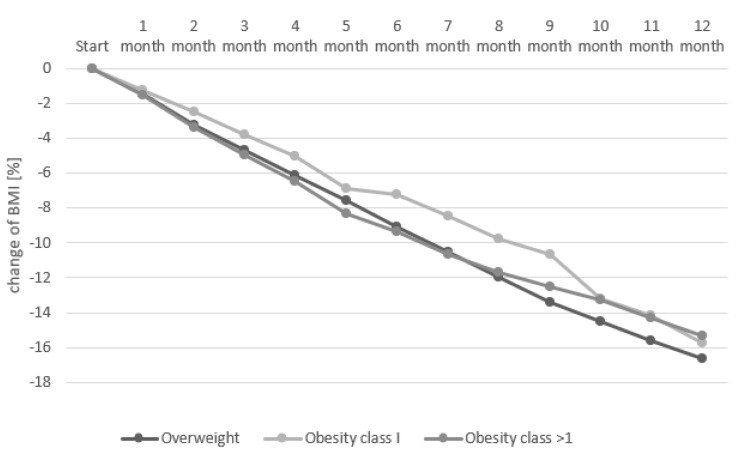
Rate of decline in BMI over 12 months in percentage (%).

**Table 1 nutrients-14-03281-t001:** General characteristics of the interventional diet.

Variable	Value
Caloric value (%)	85% TDEE
Proteins (g/kg body mass)	1.6
Fats in total (%)	25–35% of energy
Saturated FA (%)	<5% of energy
Monosaturated FA (%)	14–26% of energy
Polysaturated FA (%)	4–6% of energy
Carbohydrates (%)	50–55% of energy
Sugars (%)	<10% of energy
Fiber (g)	30–40

TDEE—total daily energy expenditure; FA—fatty acids.

**Table 2 nutrients-14-03281-t002:** Characteristics of all individuals observed at the beginning of the intervention without grouping.

Variable	The Whole Group *n =* 400 Women *=* 210 Men = 190	
	Mean ± SD	Median (Min–Max)
Age (years)	33.42 ± 7.2	32 (19–55)
Height (m)	1.73 ± 0.09	1.73 (1.50–2.02)
Body mass (kg)	95.99 ± 17.01	95 (63–156.8)
BMI (kg/m^2^)	31.83 ± 4.77	30.9 (25.1–51.7)
BMR (kcal)	1950.1 ± 342	1911 (1323–3108)
PAL	1.49 ± 0.15	1.5 (1.2–2.0)
TDEE (kcal)	2883.9 ± 454.1	2837 (1984–4603)

BMI: body mass index; BMR: basal metabolic rate; PAL: physical activity level; TDEE: total daily energy expenditure.

**Table 3 nutrients-14-03281-t003:** Characteristics of all individuals observed at the beginning of the intervention by BMI group.

Variable	Overweight (*n =* 161) Women = 87 * Men = 74 *		Obesity Class 1 (*n =* 135) Women = 67 * Men = 68 *		Obesity Class > 1 (*n =* 104) Women = 56 * Men = 48 *		*p* * 0.71
	Mean ± SD	Median (Min–Max)	Mean ± SD	Median (Min–Max)	Mean ± SD	Median (Min–Max)	*p* **
Age (years)	31.9 ± 3.01 ^a^	30 (19–54)	34.2 ± 7.2 ^b^	32 (20–55)	34.7 ± 7.7 ^b^	33 (22–55)	0.001
Height (m)	1.74 ± 0.09	1.74 (155–202)	1.73 ± 0.09	1.75 (1.5–1.96)	1.72 ± 0.8	1.72 (1.58–1.96)	0.44
Body mass (kg)	83.9 ± 10.6 ^a^	82 (63–118)	96.1 ± 11.1 ^b^	97 (69–121.9)	114.5 ± 14.6 ^c^	115 (90.9–156.8)	0.0001
BMI (kg/m^2^)	27.6 ± 1.28 ^a^	27.7 (25.1–29.7)	31.8 ± 1.1 ^b^	31.7 (30–33.8)	38.4 ± 3.5 ^c^	37.1 (35.1–51.8)	0.0001
BMR (kcal)	1745 ± 231 ^a^	1694 (1323–2470)	1964 ± 269 ^b^	2005 (1465–2559)	2248 ± 349 ^c^	2250 (1634–3108)	0.0001
PAL	1.57 ± 0.11 ^a^	1.6 (1.3–2.0)	1.46 ± 0.13 ^b^	1.4 (1.3–2.0)	1.39 ± 0.12 ^b^	1.4 (1.2–1.8)	0.0001
TDEE (kcal)	2741 ± 422 ^a^	2654 (1984–3855)	2867 ± 407 ^b^	2842 (2101–4040)	3127 ± 462 ^c^	3101 (2112–4603)	0.0001
Trainings per week	3.18 ± 0.72	3 (1–6)	3.23 ± 0.86	3 (1–6)	3.2 ± 0.96	3 (1–7)	0.79
Training time (min)	56.4 ± 17.3	60 (30–120)	56.8 ± 18	60 (30–120)	57.3 ± 19.2	60 (30–120)	0.90
Training time per week (min)	179.8 ± 70.3	180 (60–480)	183.3 ± 78	180 (30–480)	188.8 ± 91.8	180 (30–600)	0.88

* Pearson’s Chi-square test—gender between groups. ** Kruskal–Wallis one-way analysis of variance by ranks. ^a,b,c^ Kruskal–Wallis test—difference between groups.

**Table 4 nutrients-14-03281-t004:** Changes in body weight in BMI groups over 12 months.

Body Mass	Overweight (*n =* 161)					Obesity Class 1 (*n =* 135)					Obesity Class > 1 (*n =* 104)					
	Mean ± SD (kg)	Median (Min–Max) (kg)	Change (%)	Change (kg)	*p* *	Mean ± SD (kg)	Median (Min–Max) (kg)	Change (%)	Change (kg)	*p* **	Mean ± SD (kg)	Median (Min–Max)(kg)	Change (%)	Change (kg)	*p* *	*p* **
Start	83.9 ± 10.6	82 (63–118)	-	-	-	96.1 ± 11.1	97 (69–121.9)	-	-	-	114.5 ± 14.6	115 (90.9–156.8)	-	-	-	-
In 1 month	82.2 ± 10.3	81 (62–112.5)	−2.1	−1.7	0.0001	94.3 ± 10.9	95 (68–119)	−1.9	−1.8	0.0001	111.9 ± 14.3	112 (89–151.8)	−2.3	−3	0.0001	0.54
In 2 months	80.9 ± 10.2	79.4 (60.6–107.9)	−1.6 ^a^	−1.3		92.8 ± 10.9	93 (67–117)	−1.6 ^a^	−1.5		109.8 ± 14.1	109 (87–148.2)	−1.9 ^b^	−2.1		0.02
In 3 months	79.6 ± 10	78.1 (59.3–104)	−1.6	−1.3		91.6 ± 10.9	91.7 (66–116)	−1.3	−1.2		108 ± 14	107.2 (84.7–146)	−1.6	−1.8		0.06
In 4 months	78.3± 10	77.1 (57.5–103.2)	−1.6	−1.3		90.4 ± 10.9	90.2 (65–114.8)	−1.4	−1.2		106.2 ± 14.1	105.3 (80.2–144.4)	−1.7	−1.8		0.41
In 5 months	77.1± 10.04	76 (56.2–102.6)	−1.6	−1.2		88.6 ± 11.5	89.3 (64–114)	−0.9	−0.8		104.2 ± 14.2	104.1 (80.1–142.1)	−1.9	−2		0.17
In 6 months	75.9± 10.1	75 (54–101.9)	−1.7 ^a^	−1.4		88.2 ± 11.6	88.1 (63–113)	−0.5 ^b^	−0.4		102.9 ± 14.2	101.3 (75–138.1)	−1.2 ^a^	−1.3		0.02
In 7 months	74.7± 10.2	74 (52.3–101)	−1.5	−1.2		87.1 ± 11.7	87 (62–112.1	−1.2	−1.1		101.5 ± 14.4	99.9 (72.1–137)	−1.3	−1.4		0.06
In 8 months	73.6± 10.1	73 (51.8–100.2)	−1.5 ^a^	−1.1		85.9 ± 11.7	86 (59.2–111)	−1.5 ^a^	−1.2		100.4 ± 14.3	99.1 (72–136)	−1.1 ^b^	−1.1		0.002
In 9 months	72.6 ± 10.2	72 (51–99.2)	−1.4 ^a^	−1		84.9 ± 11.8	85 (57–109.8)	−1.2 ^a^	−1		99.4 ± 14.3	98.1 (71.1–135)	−1 ^b^	−1		0.0001
In 10 months	71.6 ± 10.2	70.8 (50.2–98.1)	−1.4 ^a^	−1		82.7 ± 12.2	84.2 (54.2–109)	−2.6 ^b^	−2.2		98.5 ± 14.5	97.6 (70.5–136)	−0.9 ^c^	−0.9		0.0001
In 11 months	70.7 ± 10.3	69 (50–97)	−1.5 ^a^	−0.9		82.2 ± 12.3	84 (51.1–107)	−0.6 ^b^	−0.5		97.6 ± 14.6	97 (70–137)	−0.9 ^b^	−0.9		0.0001
In 12 months	70.1 ± 10.4	68.7 (49.8–95)	−1.1 ^a^	−0.6		81.1 ± 12.4	82.5 (50.1–108)	−1.3 ^a^	−1.1		96.8 ± 14.6	96.1 (69.2–134.5)	−0.9 ^b^	−0.8		0.002
After 12 months	70.1 ± 10.4	68.7 (49.8–95)	−16.6 ^a^	−13.8		81.1 ± 12.4	82.5 (50.1–108)	−15.7 ^b^	−15		96.8 ± 14.6	96.1 (69.2–134.5)	−15.4 ^b^	−17.7		0.0007

* Significance of the change in body weight compared to the previous month—Friedman’s rank test. ** ^a,b,c^ Kruskal–Wallis test—difference between groups.

**Table 5 nutrients-14-03281-t005:** Effectiveness of a 12-month weight reduction program as expressed in changes in BMI values.

Variable	Overweight Patients (*n =* 161)	Patients with Class 1 Obesity (*n =* 135)	Patients with Class > 1 Obesity (*n =* 104)
Number of patients with BMI < 25 at the beginning of the intervention	0		
% of patients with BMI < 25 at the beginning of the intervention	0%		
Number of patients with BMI < 25 after 12 months	139	16	1
% of patients with BMI < 25 after 12 months	86.3%	11.9%	0.9%
Number of patients who reduced BMI by 1 baseline degree	139	134	82
% of patients who reduced BMI by 1 baseline degree	86.3%	99.2%	78.8%
% of patients who reduced BMI by at least 5%	100%		
% of patients who reduced BMI by at least 10%	100%	97.7%	89.4%
% of patients who reduced BMI by at least 15%	60.8%	46.1%	44.2%
% of patients who reduced BMI by at least 20%	21.2%	18.2%	16.3%

**Table 6 nutrients-14-03281-t006:** Change in basal metabolic rate (BMR) and total metabolic rate (TMR) after 12 months of intervention.

Variable	Overweight Patients (*n =* 161)	Patients with Class 1 Obesity (*n =* 135)	Patients with Class 2 and 3 Obesity (*n =* 104)	*p* *
BMR at the beginning (kcal)	1745 ± 231	1964 ± 269	2248 ± 349	0.0001
BMR after 12 months (kcal)	1497 ± 119.6	1591.2 ± 141	1737 ± 146	
BMR change in %	−13.4%	−18.9%	−22.7%	
TMR at the beginning (kcal)	2741 ± 422	2867 ± 407	3127 ± 462	
TMR after 12 months (kcal)	2350 ± 186.8	2322 ± 205	2414 ± 202.9	
TMR change in %	−13.4%	−18.9%	−22.7%	

* Kruskal–Wallis Test.

## Data Availability

The data that support the findings of this study are available from the first author (J.W.) upon reasonable request.

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
