# Peer review of "Effectiveness of a 12-Month Online Weight Reduction Program in Cohorts with Different Baseline BMI—A Prospective Cohort Study"

_nutrients, 2022, doi:10.3390/nu14163281_

Round 1
Reviewer 1 Report
After reviewing the manuscript several times, my opinion is to reject it.I think that the conclusions of the study have been known for a long time and do not provide any new concept of clinical interest. The limitations made by the study authors themselves are a cause for rejection
Reviewer 2 Report
The authors present a good research paper.
- The relevance of the topic: Good.
- Introduction: Good.
- Methodology: Can be improved.
- Results: Good.
- Discussion: Good.
In general, the paper follows an adequate structure and correct scientific support and can be published considering some limitations. The work is interesting in the field of reduction program for people with obesity. However, there are a series of limitations that should be considered.
In the first place, carry out a review of the existing literature related to the subject, being essential to inquire into the MPDI – Nutrients journal itself, since there are papers related to its manuscript that can help to improve it. Therefore, include those references, if any, especially from the last five years. Also, recommend reading some papers related to the topic of reduction program for people with obesity.
Reduction program
González-Jurado, J. A., Suárez-Carmona, W., López, S., & Sánchez-Oliver, A. J. (2020). Changes in Lipoinflammation Markers in People with Obesity after a Concurrent Training Program: A Comparison between Men and Women. International Journal of Environmental Research and Public Health, 17(17), 6168.
Moin, T., Damschroder, L. J., AuYoung, M., Maciejewski, M. L., Havens, K., Ertl, K., ... & Richardson, C. R. (2018). Results from a trial of an online diabetes prevention program intervention. American journal of preventive medicine, 55(5), 583-591.
Schurmans, G., Caty, G., & Reychler, G. (2022). Is the Peri-Bariatric Surgery Exercise Program Effective in Adults with Obesity: a Systematic Review. Obesity Surgery, 1-24.
Wadden, T. A., Tsai, A. G., & Tronieri, J. S. (2019). A protocol to deliver intensive behavioral therapy (IBT) for obesity in primary care settings: the MODEL‐IBT program. Obesity, 27(10), 1562-1566.
Specific comments.
Title: It´s righ.
Abstract. Incorporate in the summary, a more precise sentence of the results.
In their 1. Introduction. This section presents the problem in a coherent and clear manner with the correct support of the scientific literature. However, it is convenient to update the references, since there are different works related to the subject and no mention is made, and it would even be interesting to mention the different existing works related to of reduction program for people with obesity. Also, it could be a future study of review.
In section 2. Methods. Modify the method section and incorporate the sections: Design.
- Study design. To write the design section, we recommend that you take some of the following methodologists as references.
Ato, M., López-García, J. J., & Benavente, A. (2013). A classification system for research designs in psychology. Anales de Psicología/Annals of Psychology, 29(3), 1038-1059.
Montero, I., & León, O.G. (2007). A guide for naming research studies in psychology. International Journal of Clinical and Health Psychology, 7(3), 847-862.
Also, incorporate the sections of: Sample, Variables, Procedure and Statistical Analysis (Lines 80-183).
In their section 4. Results. Summary of study data and table are correct.
In section 5. Conclusion. Differentiate the discussion of the main conclusions of the work. To do this, you must create this section. And modify the limitations of the study and locate them in said section at the end. Also, they must be direct, and highlight the main contributions of the study.
In section 6. References. They should be reviewed and updated according to the publication standards.
With Kind Regards,
Reviewer 3 Report
Effectiveness of a 12-month Online Weight Reduction Program in Cohorts with Different Baseline BMI - A Retrospective Cohort Study
Summary: The authors looked at whether baseline bmi/weight and duration and rate of weight loss would produce differential results in a 12-month weight loss intervention study in participants with BMI >24.9. They found that overweight participants had an ~17% weight reduction, obese class 1 16%, obese >class 1 15%. When viewed in absolute values, the Obese >1 class had the most weight loss, but in percentage, the overweight group had the greatest loss.
The paper has strengths – it includes an international cohort, has a large sample, and was tracked for a year. I hope my comments below help improve the manuscript.
General Comment: Some of the sentences were awkwardly worded or had minor grammatical errors. I would go back and address those minor errors.
· Introduction: “Celem niniejszego artykułu jest odpowiedzieć na pytanie czy osoby chcące zredu- kować swoją masę ciała na przestrzeni 12 miesięcy mogą osiągnąć różne wyniki zależnie od tego jakie mają wyjściowe BMI oraz czy tempo spadku masy ciała jest stałe na przestrzeni poszczególnych miesięcy. W tym celu skupiliśmy się na wyrażeniu spadku masy ciała w kilogramach, czyli wartościach bezwzględnych oraz w procencie wyjściowej masy ciała, czyli w wartościach względnych.”
o The last paragraph of the introduction was not in English. Was this supposed to be in the manuscript?
· Introduction: The authors did an excellent job of explaining the important of weight reduction on attenuating chronic disease risk and how the recent COVID pandemic may have exacerbated rates of overweight and obesity. The authors explain that use of BMI may not capture all the nuances of the body, including FFM, FM, and bone density changes over the lifespan. My question is why does starting weight need to be considered in weight loss studies as well as rate duration of weight loss? These are intuitive and make sense to pay attention to this, but can the authors elaborate on this? It seems this needs more background. Also, it was a bit confusing to talk about the limitations of BMI but still use BMI to create the 3 study groups. I think this also needs to be rewritten.
· Introduction: If possible, I would also add aims, research questions, and/or hypotheses to help readers follow along and understand the purpose of the study. I would model it after the section in the discussion (We chose to focus on three aspects related to reducing one's body weight.).
· Method: “excessive weight (BMI > 24.99 kg/m2) individuals with excessive weight in between”
o I’m not sure what the “excessive weight in between” means?
· Method: The respo method of weight loss sounds very good in contouring diets and plans to fit and suit the needs of patients. However, is the respo method feasible to use in research? What parameters are adjusted for with this approach? The authors mentioned that macronutrient content was pretty constant (e.g., “The proportion of carbohydrates in 127 the diet was set at 50-55% of the energy value of the diet, with sugars added to 10%. The 128 proportion of energy from fat was 25-35% of the energy value of the diet and the protein 129 supply was set at 1.6g per kg of body weight.”), and that the diets were fit for taste preferences, but what else was manipulated with this approach? My concern is that there may be other variables that may not be accounted for in the examination of weight loss.
· Method: “This included questions regarding: Anthropometric measurements were taken by patients at baseline and during the intervention [12].”
o This also doesn’t make sense. Did the authors mean that the questions included self-reported anthropometric measurements?
· Method/Results: It seems that the body of the manuscript and the tables have different numeric presentations. It looks like the body uses decimal points and the tables use commas. I would try to be consistent in the presentation.
· Method: It was stated that the participants were trained to take their weight measurements. Was the scale electronic (I’m assuming it was – “Each participant used a standardized balance from a manufacturer certified by the Central Office of Measures in Poland to ensure the accuracy of the measurements.”) and also were there additional tests to check whether the participants were accurately weighing themselves (e.g., taking weight measurements of participants roughly on the same day as the self-measurements to show there were no “rater bias”.)
· Results: I apologize if I missed this. Is there a way to create a trajectory model where the class groups is used as an moderator (e.g., time*group)? Based on Figure 2, it looks like the slopes are the same, so I’m wondering if the weight loss is the same BUT because their starting values are different, we see this pattern.
· Results/discussion: That leads me to my next question. It seems to make sense that individuals with more weight to lose will lose more weight BUT that those with BMI’s closer to normal BMI will reach that sooner. I think the discussion should emphasize what is new about this finding.
Round 2
Reviewer 1 Report
The manuscript has been slightly improved with corrections made by the authors. However, I stand by my first report and believe that the results of the study are not new.
Author Response
Thank you for reviewing our article, however, we disagree with the statement that our study has little clinical significance.
There are few scientific publications in the topic we are researching. Especially since the topic of the observed intervention is an online intervention aimed at weight loss
Reviewer 3 Report
I carefully read the manuscript and the responses and I'm satisfied with the changes. My only comment would be to do a final grammar and spell check to catch errors (e.g., Main Conlusions, Other Conlusions).
Author Response
Thank you for your reviews. Our article was subject to language proofreading in a professional company called TRANSLMED Publishing Group dealing with the translation of scientific articles. Therefore, we trust that our work is prepared in an appropriate and proper manner.